# CBP-QSNN: Spiking Neural Networks Quantized Using Constrained Backpropagation

## Abstract

Spiking Neural Networks (SNNs) support sparse event-based data processing at high power efficiency when implemented in event-based neuromorphic processors. However, the limited on-chip memory capacity of neuromorphic processors strictly delimits the depth and width of SNNs implemented. A direct solution is the use of quantized SNNs (QSNNs) in place of SNNs with FP32 weights. To this end, we propose a method to quantize the weights using constrained backpropagation (CBP) with the Lagrangian function (conventional loss function plus well-defined weight-constraint functions) as an objective function. This work utilizes CBP as a post-training algorithm for deep SNNs pre-trained using various state-of-the-art methods including direct training (TSSL-BP, STBP, and surrogate gradient) and DNN-to-SNN conversion (SNN-Calibration), validating CBP as a general framework for QSNNs. CBP-QSNNs highlight their high accuracy insomuch as the degradation of accuracy on CIFAR-10, DVS128 Gesture, and CIFAR10-DVS in the worst case is less than 1%. Particularly, CBP-QSNNs for SNN-Calibration-pretrained SNNs on CIFAR-100 highlight an unexpected large increase in accuracy by 3.72% while using small weight-memory (3.5% of the FP32 case).

## 1 Introduction

Spiking Neural Networks (SNNs) are time-dependent models with spiking neurons whose dynamics in conjunction with synaptic current dynamics constitutes the rich dynamics of SNNs (Jeong, 2018). Deep SNNs are clearly distinguished from deep neural networks (DNNs) such that (i) presynaptic spiking neurons send out 1-bit data (spikes a.k.a. events) to their postsynaptic neurons unlike the nodes sending out real-valued activation values to the nodes in the next layer in a DNN and (ii) SNN operations are based on asynchronous sparse spikes unlike DNNs based on layerwise synchronous activation calculations (Jeong, 2018; Pfeiffer & Pfeil, 2018). These distinct features endow SNNs with high power efficiency given minimum data movements and high sparsity in operations. Yet, SNNs leverage the efficiency only when implemented in neuromorphic processors that supports event-based operations.

Neuromorphic processor design technologies are diverse, e.g., mixed analog/digital circuits (Merolla et al., 2014a; Moradi et al., 2018; Neckar et al., 2019), and fully digital circuits (Merolla et al., 2014b; Davies et al., 2018; Frenkel et al., 2018; Kornijcuk et al., 2019). Albeit diverse, all designs commonly suffer from their limited on-chip memory (SRAM) capacity. The on-chip memory is mainly assigned to neurons (state variables and hyperparameters), synapses (weights, state variables, and hyperparameters), and event-router (lookup tables). The largest portion of on-chip memory is dedicated to synaptic weights given a significant number of synapses in a deep SNN. Additionally, most neuromorphic processors hardly allow weight-reuse for convolutional SNNs because they are designed for dense SNNs. Although some compilers for weight-reuse, e.g., NXTF for Loihi (Rueckauer et al., 2021), the weight-reuse rate is still far below the ideal rate. Consequently, the limited on-chip memory capacity strictly limits the size (depth and width) of SNNs implementable in neuromorphic processors.

Considering the limitation of on-chip memory capacity, attempts to reduce the use of synaptic weight-memory have been made, which include unstructured SNN pruning (Neftci et al., 2016; Rathi et al., 2019; Martinelli et al., 2020; Chen et al., 2021; Deng et al., 2021; Kim et al., 2022; Chen et al., 2022) and weight-quantization (Rueckauer et al., 2017; Yousefzadeh et al., 2018; Srini-

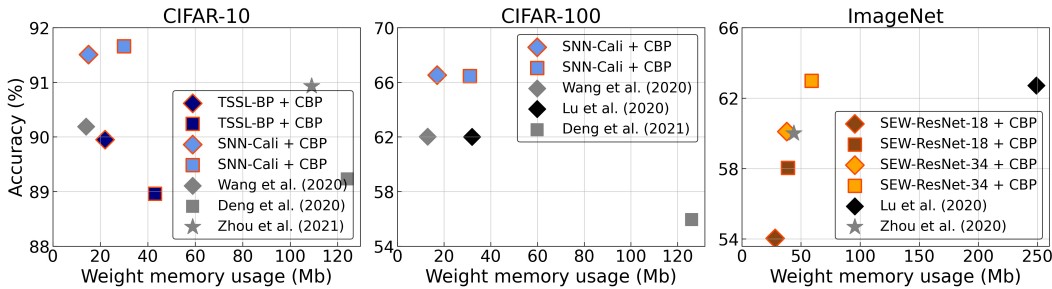

Figure 1: Accuracy and weight-memory usage of CBP-QSNNs on CIFAR-10/100 and ImageNet. The diamond, square, and star symbols denote binary, ternary, and int2 weight precision, respectively.

vasan & Roy, 2019; Lu & Sengupta, 2020; Kim et al., 2021; Deng et al., 2021; Eshraghian et al., 2022). Although unstructured SNN pruning can increase the sparsity of weight matrices, it hardly reduces the memory usage unless low precision data formats are used for such null weights, which, however, causes additional significant complexity in computation. Weight-quantization to lower resolution than the full-precision (FP32) can largely reduce weight-memory usage. Several methods proposed to date, however, are limited to (i) particular learning algorithms, e.g., spike timing-dependent plasticity (STDP) (Yousefzadeh et al., 2018; Srinivasan & Roy, 2019), event-based random backprop (eRBP) (Kim et al., 2021), rate code-based spatio-temporal backprop (Deng et al., 2021), or (ii) image datasets (e.g., MNIST, FashionMNIST, CIFAR-10/100, ImageNet) for the conversion of quantized DNNs to quantized SNNs (QSNNs) (Rueckauer et al., 2017; Wang et al., 2020; Lu & Sengupta, 2020).

Generally, deep SNNs learn optimal weights using three distinct methods: direct training using backprop based on (i) rate code (Wu et al., 2018; Shrestha & Orchard, 2018; Wu et al., 2019; Fang et al., 2021a;b; Zheng et al., 2021) and (ii) temporal code (Zhang & Li, 2020; Yang et al., 2021; Zhou et al., 2021), and (iii) DNN-to-SNN conversion (Rueckauer et al., 2017; Sengupta et al., 2019; Han et al., 2020; Deng & Gu, 2020; Li et al., 2021). Note that the last method covers SNNs on image (static rather than event-based) datasets only. Regarding this diversity in learning methods and datasets, a general weight-quantization framework can resolve the difficulty in weight-quantization. To this end, we propose a weight-quantization method based on constrained backpropagation (CBP) that uses Lagrangian functions (conventional loss function plus weight-constrain functions, each of which given a Lagrange multiplier) as objective functions (Kim & Jeong, 2021). CBP offers a general weight-quantization framework given that any constraints on weight quantization (e.g., binary, ternary, etc.) can easily be applied to the weight-constraint functions in the Lagrangian function. CBP is a post-training method so that SNNs that learned real-valued weights using different pre-training methods can be post-trained using the Lagrangian functions with the same loss functions as for the pre-training. The main contributions of our work are as follows:

- We validate CBP as a general framework for QSNNs by successfully binary- and ternary-quantizing deep SNNs (with various topologies) pre-trained on various datasets (CIFAR-10/100, ImageNet, DVS128 Gesture, CIFAR10-DVS) using representative (i) rate code-based backprop algorithms with surrogate gradients (Wu et al., 2018; Fang et al., 2021a), (ii) temporal code-based backprop algorithm (Zhang & Li, 2020), and (iii) DNN-to-SNN conversion (Li et al., 2021).

- We propose a surrogate loss function to quantize SNNs that learned real-valued weights using the DNN-to-SNN conversion method and report surprisingly high accuracy, particularly, on CIFAR-100, which exceeds the accuracy of the real-valued SNNs by more than 3%.

- We analyse the accuracy and weight-memory efficiency of CBP-QSNNs on various datasets in comparison with previous methods, which highlights the state-of-the-art (SOTA) accuracy with weight-memory usage similar to or lower than the previous methods as shown in Figure 1.

## 2 RELATED WORK

Simple methods for weight-quantization include learning uniformly quantized weights using straight through estimators (STEs) with surrogate gradients (Zhou et al., 2021; Qiao et al., 2021; Eshraghian et al., 2022). This quantization method inevitably causes large noise that hinders the loss from attaining its minima. Zhou et al. (2021) overcame the difficulty by using a novel neuron model with a differential phase-domain integrator, whereas Eshraghian et al. (2022) overcame it by periodically boosting the learning rate.

Several methods to convert binarized DNNs to QSNNs have been proposed. Rueckauer et al. (2017) proposed a method to convert BinaryConnect (Courbariaux et al., 2015) and binarized neural networks (Courbariaux et al., 2016) to binary QSNNs, whereas Lu & Sengupta (2020) to binary weight networks (Rastegari et al., 2016) to binary QSNNs. Unlike direct training methods, such conversion methods do not suffer from the issue arising from non-differentiable spiking functions, but such binary SNNs are incapable of processing event-based datasets.

There exist weight-quantization methods tailored to particular learning algorithms. Yousefzadeh et al. (2018) proposed the probabilistic update of binary weights using STDP for shallow dense SNNs. Srinivasan & Roy (2019) introduced ReStoCNet with one convolutional layer with binary kernels and one dense layer; the convolutional layer is trained using a probabilistic STDP model with binary weights, whereas the dense layer using backprop. These quantization methods suffer from the inherent drawback of STDP, i.e., its limited scalability, and thus limited learning capacity. Kim et al. (2021) proposed the eWB algorithm based on the Lagrangian function as an objective function, which is tailored to eRBP. Given the limited scalability of eRBP, eWB is also limited to shallow SNNs on datasets of low complexity. Deng et al. (2021) proposed an augmented Lagrangian function-based weight-quantization method, where the augmented Lagrangian function is minimized using the alternating direction method of multiplier (ADMM) (Boyd et al., 2011). This method is tailored to STBP with power-of-two and zero weights.

## 3 METHOD

### 3.1 CONSTRAINED BACKPROPAGATION

**Objective function** Constrained backpropagation (CBP) is a post-training method to apply independent arbitrary constraints to each weight by training DNNs using Lagrangian functions $\mathcal{L}$ as objective functions (Kim & Jeong, 2021). For CBP-QSNNs, we define the Lagrangian function consisting of a conventional loss functions $C$ and weight-constraint function $cs$ as follows.

$$
\begin{aligned}
\mathcal{L}(\boldsymbol{O}, \hat{\boldsymbol{O}}; \boldsymbol{W}, \boldsymbol{\lambda}) &= C(\boldsymbol{O}, \hat{\boldsymbol{O}}; \boldsymbol{W}) + \boldsymbol{\lambda}^{\mathrm{T}} \boldsymbol{cs}(\boldsymbol{W}), \\
\boldsymbol{cs}\left(\boldsymbol{W}\right) &= [cs\left(w_1\right), \dots, cs(w_{N_w})]^{\mathrm{T}}, \text{ and } \boldsymbol{\lambda} = [\lambda_1, \dots, \lambda_{N_w}]^{\mathrm{T}}.
\end{aligned}
\tag{1}
$$

where $\boldsymbol{O}$ is a set of SNN outputs at all timesteps, $\boldsymbol{O} = \{o^t\}_{t=1}^{T}$, and $\hat{\boldsymbol{O}}$ is the desired output. For temporal codes, $\hat{\boldsymbol{O}}$ is also given by a set of desired outputs at all timesteps, $\hat{\boldsymbol{O}} = \{\hat{o}^t\}_{t=1}^{T}$, i.e., desired output spike sequence, whereas, for rate codes, it is frequently given by a one-hot vector that indicates the correct label. Given that the loss function $C$ in Equation (1) should be identical to that for pre-training, the choice of types of $\boldsymbol{O}$ and $\hat{\boldsymbol{O}}$ relies on the pre-training method. The total number of weights in the SNN and the set of all $N_w$ weights are denoted by $N_w$ and $\boldsymbol{W}$, respectively. The value $\lambda_i$ for $i \in \{1, \dots, N_w\}$ is the Lagrange multiplier for weight $w_i$.

**Constraint function** In this work, we consider constraints of binary weights $\boldsymbol{Q} = \{-1, 1\}$ and ternary weights $\boldsymbol{Q} = \{-1, 0, 1\}$. To this end, we choose sawtooth-shaped constraint functions $cs$ that attain their minima ($cs = 0$) at $q_i \in \boldsymbol{Q}$, but modified by unconstrained-weight windows $ucs(w)$ in which the equation $cs = 0$ holds. The width of $ucs(w)$ is parameterized by $g (\geq 1)$ which is shared among all weights.

$$
ucs\left(w\right) = 1 - \sum_{i=1}^{n_q - 1} H\left(\frac{1}{2g}\left(q_{i+1} - q_i\right) - |w - m_i + \epsilon|\right),
$$

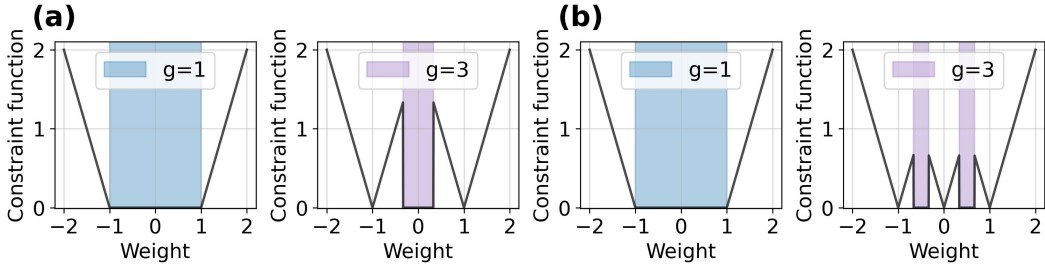

Figure 2: Constraint functions with unconstrained-weight windows (filled areas) for (a) binary- and (b) ternary-weight constraints.

where $n_q = |\boldsymbol{Q}|$, and $q_i$ and $q_{i+1}(> q_i)$ are neighboring elements in $\boldsymbol{Q}$. The median of $q_i$ and $q_{i+1}$ is denoted by $m_i(= (q_i + q_{i+1})/2)$, and $\epsilon \to 0^+$. The function $H$ denotes the Heaviside step function. When $g = 1$, $ucs = 0$ for all $w \in [\min \boldsymbol{Q}, \max \boldsymbol{Q}]$. The width of $ucs$ narrows down as $g$ increases, and it vanishes when $g \to \infty$. Examples of binary- and ternary-weight constraint functions for $g = 1$ and $3$ are shown in Figure 2.

**Parameter update**   CBP includes three types of parameters subject to update: $\boldsymbol{W}$, $\boldsymbol{\lambda}$, and $g$. The weights $\boldsymbol{W}$ and multipliers $\boldsymbol{\lambda}$ are updated using the Lagrange multiplier method (LMM). LMM with differentiable constraint functions $cs$ calculates the optimal $\boldsymbol{W}^*$ and $\boldsymbol{\lambda}^*$ such that $(\boldsymbol{W}^*, \boldsymbol{\lambda}^*) = \arg\min_{\boldsymbol{W}, \boldsymbol{\lambda}} \mathcal{L}(\boldsymbol{W}, \boldsymbol{\lambda})$. To include constraint functions that are non-differentiable at $\boldsymbol{W}^*$, we apply pseudo-LMM (Kim & Jeong, 2021) to Equation (1), which is given by

$$\nabla_{\boldsymbol{W}, \boldsymbol{\lambda}} \mathcal{L}(\boldsymbol{W}, \boldsymbol{\lambda}) = \boldsymbol{0} \Leftrightarrow \begin{cases} \nabla_{\boldsymbol{W}} C(\boldsymbol{W}) &= \boldsymbol{0}, \\ \boldsymbol{cs}(\boldsymbol{W}) &= \boldsymbol{0}. \end{cases}$$

That is, at the minima of the Lagrangian function $\mathcal{L}$, the loss function attains its minimum, and the optimal $\boldsymbol{W}$ satisfies the constrains. In search of $\boldsymbol{W}^*$, we use the basic differential multiplier method (BDMM) (Platt & Barr, 1987), which updates $\boldsymbol{W}$ and $\boldsymbol{\lambda}$ using a gradient descent and ascent method, respectively. Note that the weight update conforms to the backpropgation pipeline for the pre-training, but with the Lagrangian function $\mathcal{L}$ rather than the loss function $C$.

$$\boldsymbol{W} \leftarrow \boldsymbol{W} - \eta_W \nabla_{\boldsymbol{W}} \mathcal{L}(\boldsymbol{W}, \boldsymbol{\lambda}), \tag{2}$$

where $\eta_W$ denotes the learning rate for the weights $\boldsymbol{W}$.

Unlike the weight update, the multipliers $\boldsymbol{\lambda}$ and unconstrained-weight window parameter $g$ are updated once every learning epoch only if (i) the the sum of the Lagrangian functions over the mini-batches for a given epoch ($\mathcal{L}_{\text{sum}}$) exceeds the sum for the previous epoch ($\mathcal{L}_{\text{sum}}^{\text{pre}}$) or (ii) the update has not been executed in the past $p^{\max}$ epochs.

$$\begin{cases} \boldsymbol{\lambda} &\leftarrow \boldsymbol{\lambda} + \eta_\lambda \nabla_{\boldsymbol{\lambda}} \mathcal{L}(\boldsymbol{W}, \boldsymbol{\lambda}) = \boldsymbol{\lambda} + \eta_\lambda \boldsymbol{cs}(\boldsymbol{W}; g), \\ g &\leftarrow g + \Delta g, \text{ where } \Delta g = 1 \text{ (if } g < 10) \text{ and } 10 \text{ (if } g \geq 10). \end{cases}$$

where $\eta_\lambda$ denotes the learning rate for the Lagrange multipliers $\boldsymbol{\lambda}$. The variable $g$ is initialized to one and conditionally increases (never decreases) during training. Thus, the unconstrained-weight window $ucs$ allows rather gradual quantization of weights such that the weights outside the window (i.e., close to $q \in \boldsymbol{Q}$) are forced to be quantized (which likely increase the loss) while the weights within the window (far from $q \in \boldsymbol{Q}$) are updated to reduce the loss. CBP is elaborated in Algorithm 1 in Appendix A.1.

**Application**   CBP can works as a general weight-quantization framework because the algorithm itself is tailored to neither particular weight constraints nor particular loss functions. The loss function $C$ in Equation (1) and backpropagation pipeline for weight update can readily be chosen as per the loss function used for the pre-training. The weight-constraints can freely be chosen as per the weight resolution desired. In the following two sections, we will show applications of CBP to several distinct cases for which pre-trained SNNs with FP32 weights and/or codes are publicly available for fair baselines.

## 3.2 CBP Settings for Direct Trained SNNs

**TSSL-BP-trained conv-SNNs (temporal code)** TSSL-BP is a renowned temporal code-based error-backpropagation algorithm with data labels encoded as distinct output spike sequences $\hat{\boldsymbol{O}}(= \{\hat{\boldsymbol{o}}^t\}_{t=1}^{T}; \hat{\boldsymbol{o}}^t = \hat{\boldsymbol{s}}^t$, where $\hat{\boldsymbol{s}}^t$ denotes the desired output spike vector at timestep $t$) (Zhang & Li, 2020). Likewise, the output of the SNN $\boldsymbol{O}$ is a sequence of output spikes over the timesteps; $\boldsymbol{O} = \{\boldsymbol{s}^t\}_{t=1}^{T}$, where $\boldsymbol{s}^t$ denotes the actual output spike vector at timestep $t$. The loss function $C$ is defined using the van Rossum distance as

$$C(\boldsymbol{O}, \hat{\boldsymbol{O}}) = \frac{1}{2} \sum_{t=1}^{T} \left| (\epsilon * \hat{\boldsymbol{s}}^t) - (\epsilon * \boldsymbol{s}^t) \right|^2, \tag{3}$$

where $\epsilon$ is an exponential kernel. CBP seamlessly applies to TSSL-BP by plugging Equation (3) into Equation (1). The weights are updated while being quantized conforming to Equation (2). To this end, the backpropagation pipeline for TSSL-BP is used with the same hyper-parameters. TSSL-BP bypasses spike function gradients (which is non-differentiable) in its backward pass; instead, it addresses the gradient of potential with respect to presynaptic spike timing, which is available in a closed-form (Zhang & Li, 2020).

**STBP-trained conv-SNNs (rate code)** STBP is a representative backpropagation algorithm for SNNs with rate code (Wu et al., 2018). STBP takes the time-averaged number of output spikes as the output in response to an input sample; $\boldsymbol{O} = 1/T \sum_{t=1}^{T} \boldsymbol{s}^t$. The label for a given sample ($\hat{\boldsymbol{O}}$) is encoded as a one-hot vector. The loss function for STBP in (Wu et al., 2018) is given by

$$C(\boldsymbol{O}, \hat{\boldsymbol{O}}) = \frac{1}{2} |\hat{\boldsymbol{O}} - \boldsymbol{O}|^2.$$

STBP includes the gradients of spike functions $s$ with respect potential $u$ ($\partial s / \partial u$) in its backpropagation pipeline, which are approximated using boxcar functions which are centered at the spiking threshold $\vartheta$ (Wu et al., 2018).

$$\frac{\partial s}{\partial u} = \begin{cases} 1 & \text{if } |u - \vartheta| < a, \\ 0 & \text{otherwise,} \end{cases}$$

where $a$ is a positive constant.

**BP-trained SEW-ResNet (rate code)** SEW-ResNet is an SNN-version of ResNet, which consists of residual blocks with residual connections for identity mapping (Fang et al., 2021a). The keys to SEW-ResNet are the configuration of computational blocks and introduction of suitable spike-element-wise logic functions to fit the identity mapping in SNNs. Fang et al. (2021a) deployed integrate-and-fire neurons in the output layer and considered their potential $u$ at the last timestep ($u^T$) which is normalized by the the number of total timesteps $T$, i.e., $a = u^T / T$, as the output of the SNN. The output vector is then processed by the subsequent softmax layer to use the crossentropy loss as the loss function $C$.

$$C(\boldsymbol{O}, \hat{\boldsymbol{O}}) = - \sum_{c=1}^{N} \hat{o}_c \log \left( o_c \right), \tag{4}$$

$$o_c = \frac{\exp(a_c)}{\sum_{c=1}^{N} \exp(a_c)}, \text{ where } a_c = \frac{u_c^T}{T},$$

SEW-ResNet also involves the gradients of spike functions with potential $u$, which are approximated to the arctan-based surrogate gradient function (Fang et al., 2021b).

$$\frac{\partial s}{\partial u} = \frac{\alpha}{1 + (\alpha \pi (u - \vartheta))^2}. \tag{5}$$

where $\alpha$ is a positive constant.

### 3.3 CBP Settings for SNNs Converted From DNNs

DNN-to-SNN conversion methods often apply to indirectly train SNNs (Rueckauer et al., 2017; Sengupta et al., 2019; Han et al., 2020; Deng & Gu, 2020; Li et al., 2021). Among them, we chose the SNN-Calibration algorithm (Li et al., 2021) given its SOTA accuracy on various static datasets. SNN-Calibration minimizes accuracy-degradation for the limited number of timesteps by attaining the optimal layer-wise threshold ($\vartheta^{(l)}$ for a layer $l$) that minimizes the discrepancy in layerwise output between the SNN and its DNN counterpart. Additionally, (i) bias for each neuron and/or (ii) initial potentials and weights are tweaked to minimize the layerwise output error.

The SNN-Calibration-pretrained SNN utilizes spiking neurons that output real-valued spikes $\vartheta^{(l)}$, which is not suitable for CBP. Therefore, we first rescale the weights $\boldsymbol{W}^{(l)}$, biases $\boldsymbol{b}^{(l)}$, and threshold $\vartheta^{(l)}$ in the pre-trained SNN as follows.

$$\forall l, \boldsymbol{W}^{(l)} \leftarrow \boldsymbol{W}^{(l)} \frac{\vartheta^{(l-1)}}{\vartheta^{(l)}}, \quad \boldsymbol{b}^{(l)} \leftarrow \frac{\boldsymbol{b}^{(l)}}{\vartheta^{(l)}}, \quad \vartheta^{(l)} \leftarrow 1. \tag{6}$$

This rescaling allows the spiking neuron to output integer binary spikes.

Given that conversion methods do not use loss functions, we employ surrogate loss functions $C$ that are consistent with the neural code used in the pre-trained SNN. SNN-Calibration uses the time-averaged number of spikes (spike-rate) as neural input and output, so that rate code-based loss functions are appropriate surrogate loss functions as for the previous two examples. Here, we chose a cross-entropy loss function with a softmax layer as a surrogate loss function $C$ as for SEW-ResNet in Equation (4). Similar to SEW-ResNet, we also used the arctan surrogate gradient function in Equation (5) with $\alpha = 1$. However, not only weights $\boldsymbol{W}$ but also biases $\boldsymbol{b}$ were subject to post-learning. Note that the biases $\boldsymbol{b}$ were updated using the loss function $C$ as an objective function using the learning rate $\eta_b (= \eta_W)$. Stochastic gradient descent was used to learn optimal biases $\boldsymbol{b}^*$ as for the weights. CBP for this case is detailed in the pseudocode in Algorithm 2 in Appendix A.1.

## 4 Experiments

We identified the classification efficacy and weight-memory efficiency of CBP-QSNNs (with binary $\boldsymbol{Q} = \{-1, 1\}$ and ternary weight $\boldsymbol{Q} = \{-1, 0, 1\}$ constraints) for (i) TSSL-BP-trained conv-SNNs, (ii) STBP-trained conv-SNNs, (iii) BP-trained SEW-ResNet, and (iv) SNN-Calibration-trained SNNs with the settings elaborated in Secion 3. We used total five datasets: three image datasets (CIFAR-10/100 (Krizhevsky, 2009) and ImageNet (Russakovsky et al., 2015)) and two event-based datasets (DVS 128 Gesture (Amir et al., 2017) and CIFAR10-DVS(Li et al., 2017)). Note that we used publicly available pre-trained models and/or codes only for fair baselines. The input pre-processing and encoding conformed to the methods for the pre-training without modifications, which are detailed in the corresponding publications. CBP needs two optimizers (one for weight update and the other for Lagrange multiplier update). For the weight update, we used the same optimizer as for the weight pre-training, which are listed in Table 4 in Appendix A.2. Hereafter, we refer to CBP-QSNN for SNN pre-trained using a particular algorithm as CBP-QSNN-(pre-training algorithm), e.g., CBP-QSNN-(TSSL-BP) and CBP-QSNN-(SNN-Calibration), except CBP-QSNN-(SEW-ResNet) whose pre-training algorithm does not have a compact name.

We used layerwise scaling factors $c^{(l)}$ for layers $l$, which are given by $c^{(l)} = ||\boldsymbol{W}^{(l)}||_1/n^{(l)}$, where $\boldsymbol{W}^{(l)}$ and $n^{(l)}$ are the weight matrix for the layer $l$ and the number of elements in $\boldsymbol{W}^{(l)}$, respectively. For CBP-QSNN-(TSSL-BP), we used AdamW (Loshchilov & Hutter, 2019) as a Lagrange multiplier optimizer, whereas Adam (Kingma & Ba, 2014) for the others. The hyper-parameters for CBP-QSNNs are listed in Table 4 in Appendix A.2. Note that the weights for the first and last layers were not subject to quantization as for common QDNNs (Rastegari et al., 2016; Li et al., 2016). All statistic data were acquired from three trials to avoid the random seed effect except on ImageNet. The CBP algorithm was implemented in Pytorch (Paszke et al., 2019) on a GPU workstation (RTX A6000).

Table 1: Classification accuracy and weight-memory usage for CBP-QSNNs that were pre-trained using direct training methods.

| Network | Algorithm | Weight precision | Avg. (Best) accuracy | weight-memory usage |
|---|---|---|---|---|
| **CIFAR-10** | | | | |
| AlexNet | TSSL-BP | FP32 | 89.00[†](89.22[‡]) | 677Mb |
| | **CBP** | **Binary** | **89.45±0.37 (89.95)** | **21.6Mb** |
| | | **Ternary** | **88.92±0.03 (88.96)** | **42.7Mb** |
| 7Conv, 3FC | STBP | FP32 | 90.26[†](89.53[‡]) | 1.99Gb |
| | **CBP** | **Binary** | **89.51±0.05 (89.59)** | **62.4Mb** |
| | | **Ternary** | **89.58±0.08 (89.67)** | **124Mb** |
| **CIFAR-100** | | | | |
| 7Conv, 3FC | STBP | FP32 | 62.25 | 1.99Gb |
| | **CBP** | **Binary** | **60.81±0.01 (60.83)** | **63.8Mb** |
| | | **Ternary** | **60.85±0.18 (61.09)** | **126Mb** |
| **ImageNet** | | | | |
| SEW-ResNet-18 | Arctan gradient | FP32 | 62.81[†](63.18[‡]) | 374Mb |
| | **CBP** | **Binary** | **54.34** | **27.8Mb** |
| | | **Ternary** | **58.04** | **39.0Mb** |
| SEW-ResNet-34 | Arctan gradient | FP32 | 66.65[†](67.04[‡]) | 697Mb |
| | **CBP** | **Binary** | **60.10** | **37.9Mb** |
| | | **Ternary** | **62.98** | **59.2Mb** |
| **DVS128 Gesture** | | | | |
| 7B-Net | Arctan gradient | FP32 | 97.92[†](97.92[‡]) | 4.16Mb |
| | **CBP** | **Binary** | **97.11±0.33 (97.57)** | **159Kb** |
| | | **Ternary** | **97.68±0.16 (97.92)** | **288Kb** |
| **CIFAR10-DVS** | | | | |
| Wide-7B-Net | Arctan gradient | FP32 | 74.7[†](74.4[‡]) | 40.2Mb |
| | **CBP** | **Binary** | **73.89±0.62 (74.7)** | **1.33Mb** |
| | | **Ternary** | **74.83±0.46 (75.3)** | **2.58Mb** |

[†]Reproduced result [‡]Reported result

To measure the success in weight quantization, we introduce the constraint-failure score (CFS) which is the average weight-constraint function over all weights included in the SNN.

$$\text{CFS}(\boldsymbol{W}) = \frac{1}{N_w} \sum_{i=1}^{N_w} cs(w_i),$$

where $N_w$ indicates the total number of weights in the SNN. During quantization, CFS keeps decreasing toward zero because the constraint functions $cs$ attain their minima ($cs = 0$) when the weights are fully quantized. Few weights that have not been quantized until the last CBP epoch were forced to be quantized to their nearest quantized weights.

### 4.1 CBP-QSNNs WITH TRUE LOSS FUNCTIONS

**CBP-QSNN-(TSSL-BP)** We applied CBP with binary- and ternary-weight constraints to AlexNet (96C3-256C3-MP2-384C3-MP2-384C3-256C3-FC1024-FC1024-FC10) pre-trained on CIFAR-10 using TSSL-BP. The results shown in Table 1 highlight a surprising increase in classification accuracy by 0.45% when binarized, using merely a weight-memory of 21.6Mb, i.e., 3.19% of that for the FP32 weight case. The test accuracy and CFS curves are plotted in Figure 3 in Appendix A.3. Additionally, the weight distributions sampled at a few epochs are plotted in Figure 3, which show gradual weight-quantization.

**CBP-QSNN-STBP** We considered the deep SNN (7Conv,3FC) (Deng et al., 2021) which was pre-trained on CIFAR-10 using STBP, yielding an accuracy of 90.26 with FP32 weights (Table 1). The corresponding CBP-QSNN-STBPs with binary- and ternary-weight constraints show accuracy degradations of 0.75% and 0.68% while reducing weight-memory usage by 96.9% and 93.8%, respectively. The test accuracy and CFS curves, and weight distributions sampled at four learning

Table 2: Classification accuracy and weight-memory usage for CBP-QSNN-(SNN-Calibration).

| Network | Algorithm | Weight precision | Avg. (Best) accuracy | weight-memory usage |
|---------|-----------|------------------|---------------------|---------------------|
| **CIFAR-10** | | | | |
| VGG-16 | SNN-Calibration | FP32 | 91.04† (91.82‡) | 471Mb |
| | **CBP** | **Binary**
**Ternary** | **91.40±0.09 (91.51)**
**91.49±0.13 (91.66)** | **15.1Mb**
**29.8Mb** |
| **CIFAR-100** | | | | |
| VGG-16 | SNN-Calibration | FP32 | 62.60† (64.53‡) | 473Mb |
| | **CBP** | **Binary**
**Ternary** | **66.32±0.16 (66.53)**
**66.37±0.44 (66.46)** | **16.6Mb**
**31.2Mb** |

†Reproduced result ‡Reported result

epochs are shown in Figure 4 in Appendix A.3. Despite the non-availability of baseline performance in public, we also applied CBP to the same SNN on CIFAR-100 for a comparison with another weight-quantization method, which will be addressed in Section 4.3. The results are also listed in Table 1, and the quantization behaviors are plotted in Figure 5 in Appendix A.3.

**CBP-QSNN-(SEW-ResNet)** We addressed the pre-trained SEW-ResNet-18 and SEW-ResNet-34 on ImageNet, which use ADD operations as element-wise operations in the SEW blocks. CBP applied to SEW-ResNet-18 and SEW-ResNet-34 with binary- and ternary-weight constraints yielded the accuracy and weight-memory usage shown in Table 1. The quantization curves are plotted in Figures 6 and 7 in Appendix A.3.

We also applied CBP to 7B-Net and Wide-7B-Net pre-trained on neuromorphic datasets (DVS128 Gesture and CIFAR10-DVS, respectively). 7B-Net is structured as 32C3-BN-PLIF-{SEW Block(32C)-MP2}*7-FC11, where BN denotes batch normalization. PLIF denotes a layer of parametric LIF neurons which use trainable membrane time constants (Fang et al., 2021b). Wide-7B-Net is structured as 64C3-BN-PLIF-{SEW Block(64C)-MP2}*4-128C3-BN-PLIF-{SEW Block(128C)-MP2}*3-FC10. The results are shown in Table 1, and the quantization curves are plotted in Figures 8 and 9 in Appendix A.3. An anomaly is seen in the ternary-weight CBP-QSNN whose accuracy is rather larger than the FP32 counterpart by 0.13%.

## 4.2 CBP-QSNNs WITH SURROGATE LOSS FUNCTIONS

**CBP-QSNN-(SNN-Calibration)** VGG-16 SNNs were CBP-quantized on CIFAR-10 and 100 with binary- and ternary-weight constraints. To acquire CBP-QSNN-(SNN-Calibration), we used the surrogate loss function and gradient function as for CBP-QSNN-(SEW-ResNet) in Equations (4) and (5), respectively. We applied weight decay (L2-regularization) with a decay rate $w_d$ of $10^{-4}$. Table 2 shows the accuracy and weight-memory usage, highlighting considerable increases in accuracy upon quantization by 0.36% (binary case) and 0.45% (ternary case) on CIFAR-10, and 3.72% (binary case) and 3.77% (ternary case) on CIFAR-100. The quantization curves and sampled weight distributions on CIFAR-10 and 100 are plotted in Figures 10 and 11 in Appendix A.3, respectively.

## 4.3 COMPARISON WITH PREVIOUS QUANTIZATION RESULTS

We finally compare CBP-QSNNs with QSNNs by previous weight-quantization methods in terms of classification accuracy and weight-memory usage on CIFAR-10/100, ImageNet, DVS128 Gesture, and CIFAR10-DVS. The comparison is shown in Table 3.
**CIFAR-10:** CBP-QSNNs exhibit outstanding accuracy and weight-memory efficiency. Particularly, CBP-QSNN-STBP with ternary weights outperforms (Deng et al., 2021) of the same network architecture by 0.57% in accuracy. Additionally, the ternary CBP-QSNN-(SNN-Calibration) of VGG-16 exceeds the SOTA accuracy of (Zhou et al., 2021) by 0.56% with a reduction in weight-memory usage by 72.7%.
**CIFAR-100:** CBP-QSNN-(SNN-Calibration) of VGG-16 outperforms the SOTA accuracy (Wang et al., 2020) by 4.3% by using an additional 3.6Mb weight-memory only. CBP-QSNN-STBP of conv-SNN (7Conv,3FC) records 60.85% in accuracy, exceeding the accuracy of (Deng et al., 2021)

Table 3: Comparison with previous works on QSNNs with binary, ternary, and int2 weight precision.

| Algorithm | Network | Weight precision / memory usage (bit) | Accuracy (%) |
|---|---|---|---|
| **CIFAR-10** | | | |
| BNN-SNN (Rueckauer et al., 2017) | 6Conv, 3FC | Binary/14.0M | 88.25 |
| BinaryConnect-SNN (Rueckauer et al., 2017) | 6Conv, 3FC | Binary/14.0M | 83.35 |
| BinaryConnect-SNN (Wang et al., 2020) | 6Conv, 3FC | Binary/14.0M | 90.19 |
| STBP + ADMM (Deng et al., 2021) | 7Conv, 3FC | Ternary/124M | 89.01 |
| Direct-training + STE (Zhou et al., 2021) | SpikingVGG16 | int2/109M | 90.93 |
| **CBP-QSNN-STBP** | **7Conv, 3FC** | **Binary/62.4M** | **89.51±0.05 (89.59)** |
| | | **Ternary/124M** | **89.58±0.08 (89.67)** |
| **CBP-QSNN-(TSSL-BP)** | **AlexNet** | **Binary/21.6M** | **89.45±0.37 (89.95)** |
| | | **Ternary/42.7M** | **88.92±0.03 (88.96)** |
| **CBP-QSNN-(SNN-Calibration)** | **VGG-16** | **Binary/15.1M** | **91.40±0.09 (91.51)** |
| | | **Ternary/29.8M** | **91.49±0.13 (91.66)** |
| **CIFAR-100** | | | |
| BinaryConnect-SNN (Wang et al., 2020) | 6Conv, 2FC | Binary/13.0M | 62.02 |
| BWN-SNN (Lu & Sengupta, 2020) | VGG-15 | Binary/32.1M | 62.00 |
| STBP + ADMM (Deng et al., 2021) | 7Conv, 3FC | Ternary[a]/126M | 55.95 |
| **CBP-QSNN-STBP** | **7Conv, 3FC** | **Binary/63.8M** | **60.81±0.01 (60.83)** |
| | | **Ternary/126M** | **60.85±0.18 (61.09)** |
| **CBP-QSNN-(SNN-Calibration)** | **VGG-16** | **Binary/16.6M** | **66.32±0.16 (66.53)** |
| | | **Ternary/31.2M** | **66.37±0.08 (66.46)** |
| **ImageNet** | | | |
| BWN-SNN (Lu & Sengupta, 2020) | VGG-15 | Binary/249M | 62.71 |
| Direct-training + STE (Zhou et al., 2021) | GoogLeNet | int2/44.5M | 60.0 |
| **CBP-QSNN-(SEW-ResNet)** | **SEW-ResNet18** | **Binary/27.8M** | **54.34** |
| | | **Ternary/39.0M** | **58.04** |
| | **SEW-ResNet34** | **Binary/37.9M** | **60.10** |
| | | **Ternary/59.2M** | **62.98** |
| **DVS128 Gesture** | | | |
| BSNN (Qiao et al., 2021) | 2Conv, 2FC | Binary/1.68M | 97.57 |
| **CBP-QSNN-(7B-Net)** | **7B-Net** | **Binary/159K** | **97.11±0.33 (97.57)** |
| | | **Ternary/288K** | **97.68±0.16 (97.92)** |
| **CIFAR10-DVS** | | | |
| BSNN Qiao et al. (2021) | 2Conv, 2FC | Binary/1.68M | 62.10 |
| **CBP-QSNN-(wide-7B-Net)** | **Wide-7B-Net** | **Binary/1.33M** | **73.89±0.62 (74.7)** |
| | | **Ternary/2.58M** | **74.83±0.46 (75.3)** |

[a]25% connection-pruned.

of the same network topology and weight precision by 4.9%. Note that the accuracy in (Deng et al., 2021) is of a 25% connection-pruned SNN.

**ImageNet:** CBP-QSNN-(SEW-ResNet34) with ternary weights outperforms the SOTA results (Lu & Sengupta, 2020) by 0.27% in accuracy with reducing weight-memory usage by 76.2%.

**DVS128 Gesture:** For DVS128 Gesture, CBP-QSNN-(7B-Net) with ternary weights highlight an accuracy improvement by 0.11% and a weight-memory reduction by 82.9% compared with (Qiao et al., 2021).

**CIFAR10-DVS:** CBP-QSNN-(Wide-7B-Net) with binary weights achieves a considerable accuracy improvement by 11.79% with a reduction in weight-memory usage by 20.8% compared with (Qiao et al., 2021).

## 5 CONCLUSION

We proposed CBP for weight-quantization in SNNs, which can readily be adjusted to particularly pre-trained SNNs and desired weight-quantization conditions. We validated CBP as a general framework for QSNNs by reporting total 18 weight-quantization cases that vary in weight bitwidth (binary and ternary), pre-training method (TSSL-BP, STBP, arctan gradient, and SNN-Calibration), network topology, and dataset (CIFRA-10/100, ImageNet, DVS128 Gesture, and CIFAR10-DVS). CBP for all datasets highlights its SOTA results in terms of classification accuracy and weight-memory usage in comparison with previous methods for QSNNs. Therefore, CBP-QSNNs are high accuracy and high memory-efficient models that are mapped onto a neuromorphic processor with strictly limited on-chip memory capacity.

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

# A APPENDIX

## A.1 PSEUDOCODE

---

**Algorithm 1:** CBP algorithm. $N$ denotes the number of training epochs in aggregate. $M$ denotes the number of mini-batches of the training set $\boldsymbol{Tr}$. The function minibatch($\boldsymbol{Tr}$) samples a mini-batch of training data and their targets from $\boldsymbol{Tr}$. The function SNN returns output $\boldsymbol{O}$ for $\boldsymbol{x}$.

---

**Input:** Pre-trained weights $\boldsymbol{W}$ in 32FP
**Output:** Quantized weights $\boldsymbol{W}_q$
Initialization such that $\boldsymbol{\lambda} \leftarrow \boldsymbol{0}, p \leftarrow 0, g \leftarrow 1$;
Initial update of $\boldsymbol{\lambda}$;
**for** $epoch = 1$ **to** $N$ **do**
  $\mathcal{L}_{\text{sum}} \leftarrow 0$;
  /* Update of weight $\boldsymbol{W}$                                          */
  **for** $i = 1$ **to** $M$ **do**
    $\boldsymbol{x}^{(i)}, \hat{\boldsymbol{x}}^{(i)} \leftarrow$ minibatch($\boldsymbol{Tr}$);
    $\boldsymbol{O} \leftarrow \text{SNN}(\boldsymbol{x}^{(i)}; \boldsymbol{W})$;
    $\mathcal{L} \leftarrow C(\boldsymbol{O}, \hat{\boldsymbol{O}}; \boldsymbol{W}) + \boldsymbol{\lambda}^{\text{T}} \boldsymbol{cs}(\boldsymbol{W}; g)$;
    $\mathcal{L}_{sum} \leftarrow \mathcal{L}_{sum} + \mathcal{L}$;
    $\boldsymbol{W} \leftarrow \text{clip}(\boldsymbol{W} - \eta_W \nabla_{\boldsymbol{W}} \mathcal{L})$;
  **end**
  /* Update of window variable $g$ and Lagrange multiplier $\boldsymbol{\lambda}$   */
  $p \leftarrow p + 1$;
  **if** $\mathcal{L}_{sum} \geq \mathcal{L}_{sum}^{pre}$ or $p = p_{max}$ **then**
    $g \leftarrow g + \Delta g$;
    $\boldsymbol{\lambda} \leftarrow \boldsymbol{\lambda} + \eta_\lambda \boldsymbol{cs}(\boldsymbol{W}; g)$;
    $p \leftarrow 0$;
    $\mathcal{L}_{\text{sum}}^{\text{pre}} \leftarrow \mathcal{L}_{\text{sum}}^{\text{max}}$;
  **else**
    $\mathcal{L}_{\text{sum}}^{\text{pre}} \leftarrow \mathcal{L}_{\text{sum}}$;
  **end**
**end**

---

---

**Algorithm 2:** CBP for SNN-Calibration

---

**Input:** Pre-trained weights $\boldsymbol{W}$ and biases $\boldsymbol{b}$ in FP32
**Output:** Quantized weights $\boldsymbol{W}_q$ and optimal biases $\boldsymbol{b}^*$
Initialization such that $\forall l, \boldsymbol{W}^{(l)} \leftarrow \boldsymbol{W}^{(l)}\vartheta^{(l-1)}/\vartheta^{(l)}, \boldsymbol{b}^{(l)} \leftarrow \boldsymbol{b}^{(l)}/\vartheta^{(l)}, \vartheta^{(l)} \leftarrow 1$;
Initialization such that $\boldsymbol{\lambda} \leftarrow \boldsymbol{0}, p \leftarrow 0, g \leftarrow 1$;
Initial update of $\boldsymbol{\lambda}$;
**for** $epoch = 1$ **to** $N$ **do**
    $\mathcal{L}_{\text{sum}} \leftarrow 0$;
    /* Update of weights $\boldsymbol{W}$ and biases $\boldsymbol{b}$                    */
    **for** $i = 1$ **to** $M$ **do**
        $\boldsymbol{x}^{(i)} \leftarrow \text{minibatch}(\boldsymbol{Tr})$;
        $\boldsymbol{O} \leftarrow \text{SNN}(\boldsymbol{x}^{(i)}; \boldsymbol{W})$;
        $\mathcal{L} \leftarrow C(\boldsymbol{O}, \hat{\boldsymbol{O}}; \boldsymbol{W}, \boldsymbol{b}) + \boldsymbol{\lambda}^{\text{T}}\boldsymbol{cs}(\boldsymbol{W}; g)$;
        $\mathcal{L}_{sum} \leftarrow \mathcal{L}_{sum} + \mathcal{L}$;
        $\boldsymbol{W} \leftarrow \text{clip}(\boldsymbol{W} - \eta_W \nabla_{\boldsymbol{W}}\mathcal{L})$;
        $\boldsymbol{b} \leftarrow \text{clip}(\boldsymbol{b} - \eta_b \nabla_{\boldsymbol{b}}C)$;
    **end**
    /* Update of window variable $g$ and Lagrange multiplier $\boldsymbol{\lambda}$    */
    $p \leftarrow p + 1$;
    **if** $\mathcal{L}_{sum} \geq \mathcal{L}_{sum}^{pre}$ *or* $p = p_{max}$ **then**
        $g \leftarrow g + \Delta g$;
        $\boldsymbol{\lambda} \leftarrow \boldsymbol{\lambda} + \eta_\lambda \boldsymbol{cs}(\boldsymbol{W}; g)$;
        $p \leftarrow 0$;
        $\mathcal{L}_{\text{sum}}^{\text{pre}} \leftarrow \mathcal{L}_{\text{sum}}^{\text{max}}$;
    **else**
        $\mathcal{L}_{\text{sum}}^{\text{pre}} \leftarrow \mathcal{L}_{\text{sum}}$;
    **end**
**end**

---

## A.2 HYPER-PARAMETERS FOR CBP

Table 4: Hyper-parameters used in our experiments. The learning rates for weights $\boldsymbol{W}$, Lagrange multipliers $\boldsymbol{\lambda}$, and biases $\boldsymbol{b}$ are denoted by $\eta_w$, $\eta_\lambda$, and $\eta_b$, respectively. The weight-decay (L2 regularization) is denoted by $w_d$.

| Algorithm | Dataset | $\eta_w$ | $w_d$ | $\eta_b$ | $\eta_\lambda$ | Batch size | Epoch | Steps | Optimizer weight | Optimizer multiplier |
|---|---|---|---|---|---|---|---|---|---|---|
| TSSL-BP | CIFAR-10 | 2e-4 | - | - | 1e-4 | 50 | 200 | 5 | AdamW | AdamW |
| STBP | CIFAR-10 CIFAR-100 | 5e-3 | - | - | 5e-4 | 50 | 100 | 8 | SGD | Adam |
| SEW-ResNet | DVS128 Gesture CIFAR10-DVS ImageNet | 0.1 | - | - | 0.01 | 16 16 32 | 200 64 100 | 16 16 4 | SGD | Adam |
| SNN-Calib | CIFAR-10 CIFAR-100 | 0.01 | 1e-4 | 0.01 | 1e-3 | 128 | 100 | 32 | SGD | Adam |

## A.3 Extra data

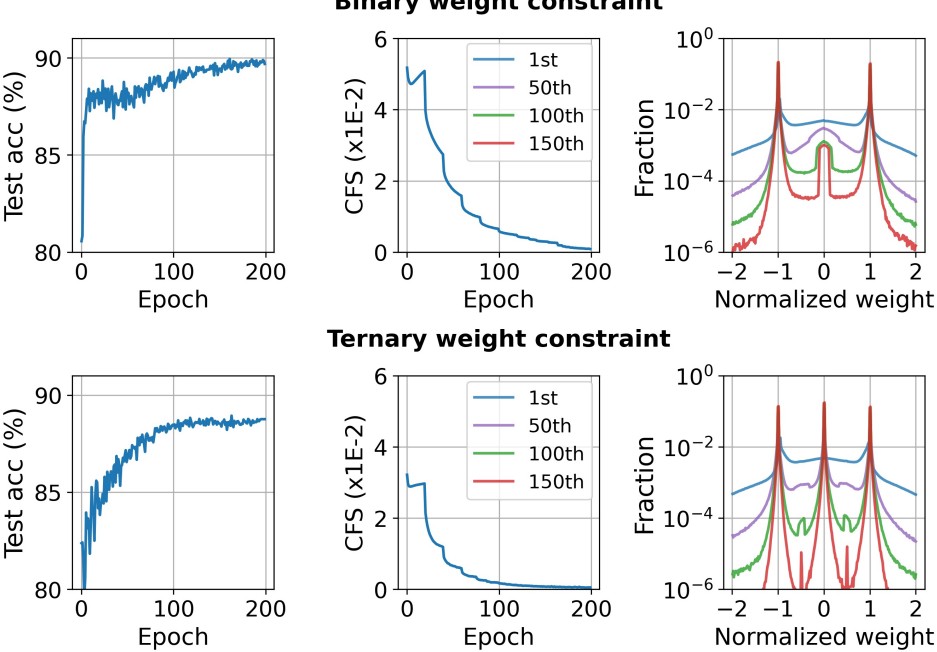

Figure 3: CBP-QSNN-(TSSL-BP) on CIFAR-10. The QSNN topology is of AlexNet.

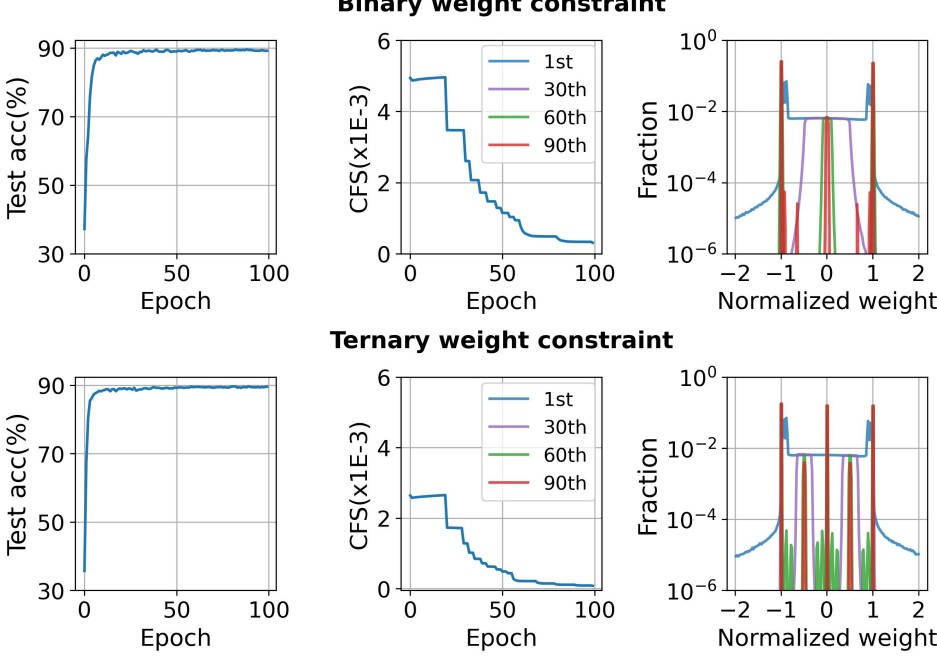

Figure 4: CBP-QSNN-STBP on CIFAR-10. The QSNN topology is of a deep SNN (7Conv and 3FC).

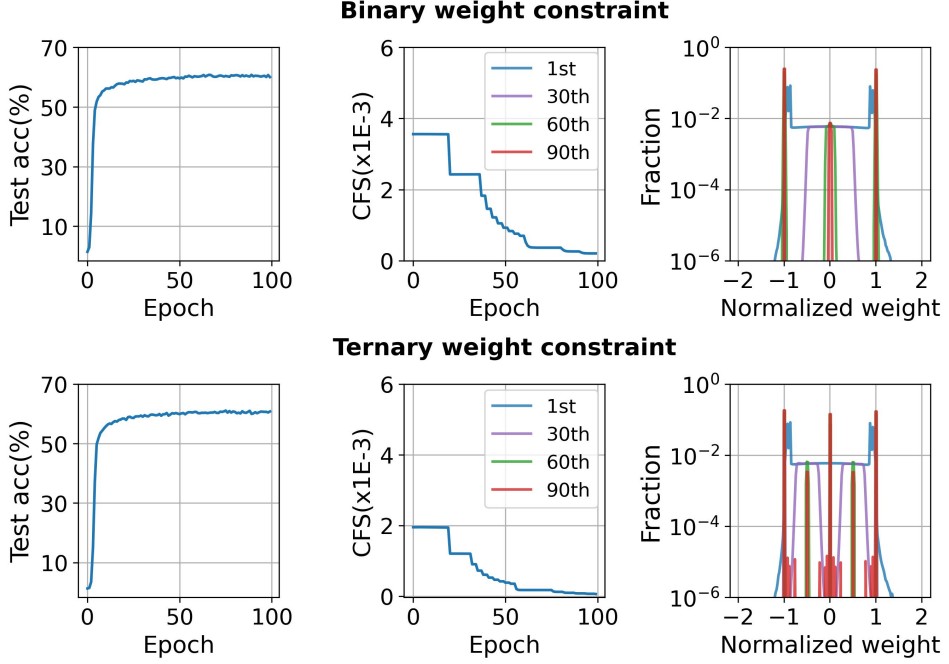

Figure 5: CBP-QSNN-STBP on CIFAR-100. The QSNN topology is of a deep SNN (7Conv and 3FC).

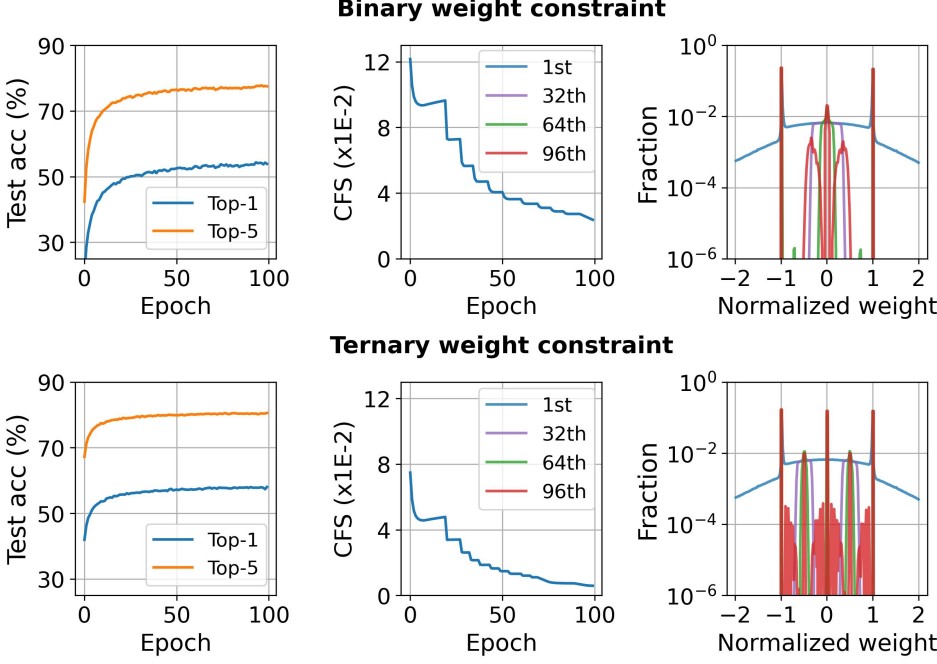

Figure 6: CBP-QSNN-(SEW-ResNet-18) on ImageNet.

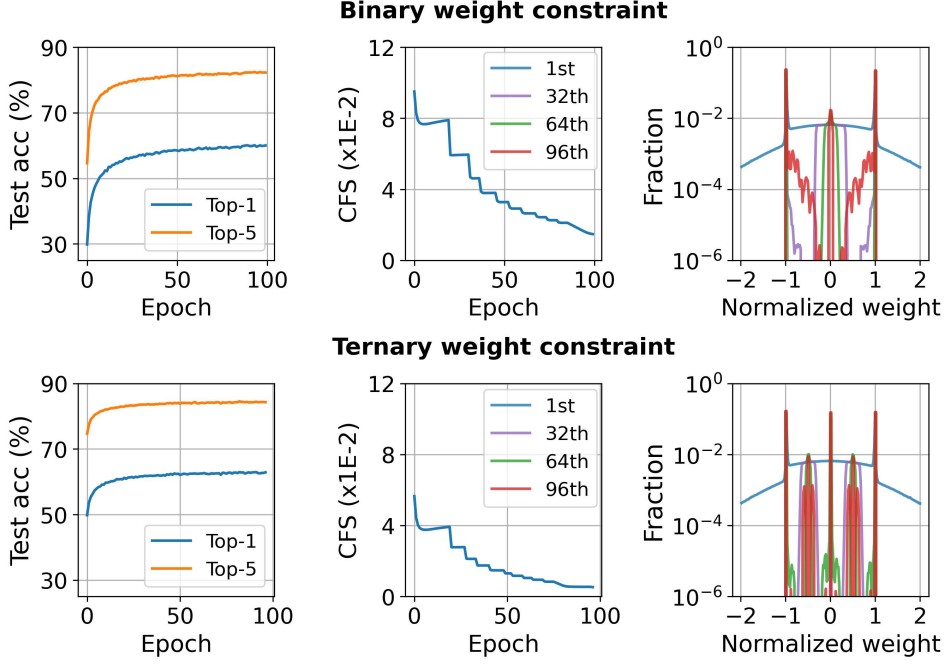

Figure 7: CBP-QSNN-(SEW-ResNet-34) on ImageNet.

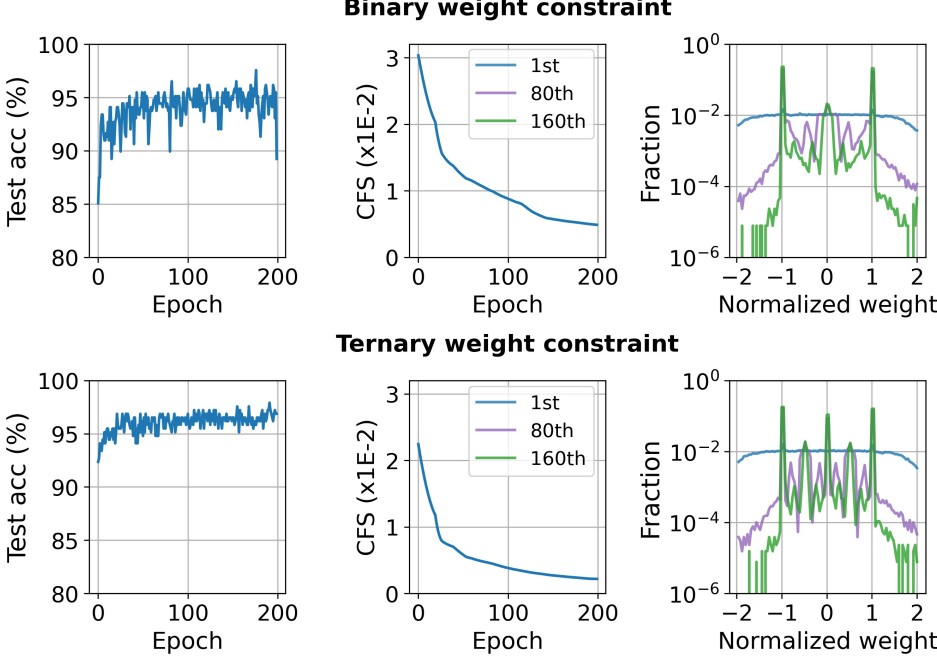

Figure 8: CBP-QSNN-(7B-Net) on DVS128 Gesture.

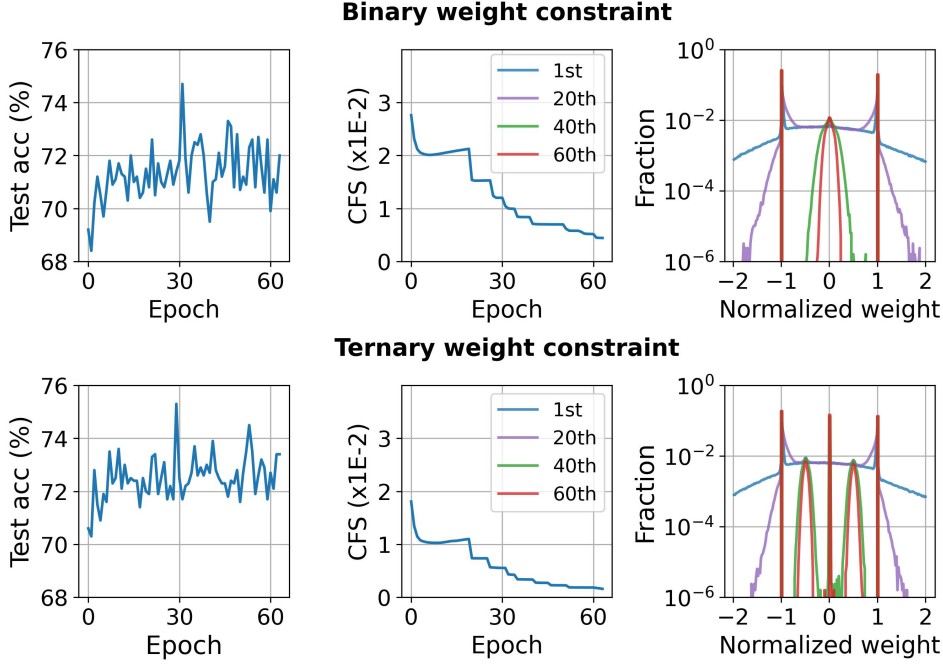

Figure 9: CBP-QSNN-(Wide-7B-Net) on CIFAR10-DVS.

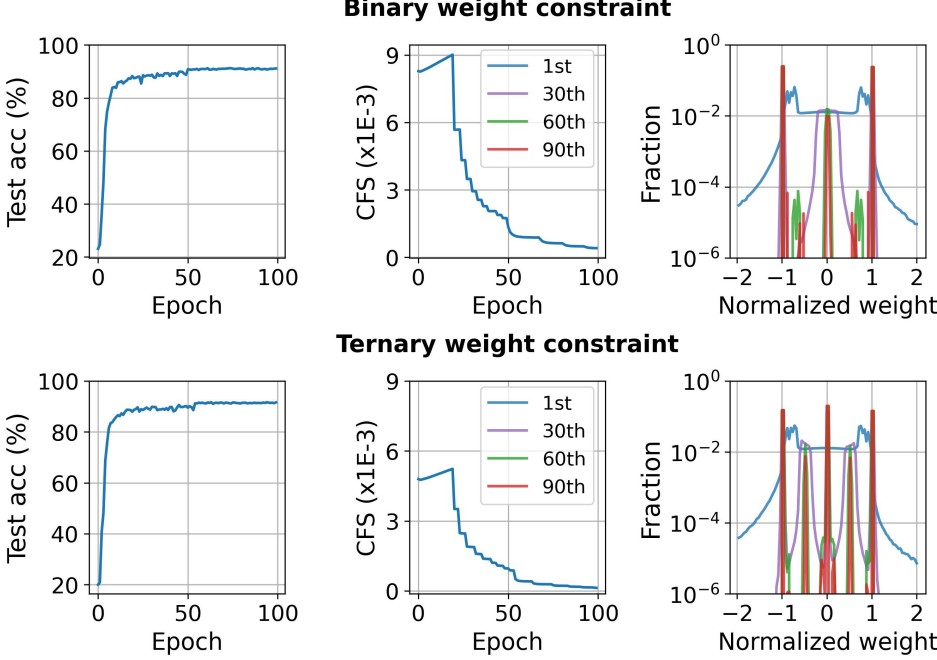

Figure 10: CBP-QSNN-(SNN-calibration) on CIFAR-10. The QSNN topology is of VGG-16.

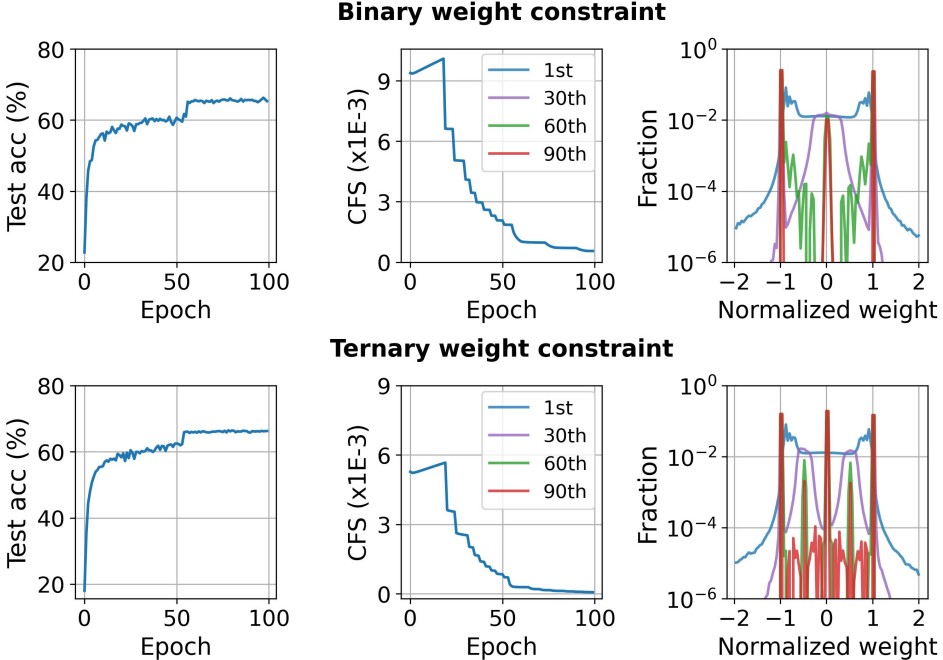

Figure 11: CBP-QSNN-(SNN-calibration) on CIFAR-100. The QSNN topology is of VGG-16.

