# OpenReview forum: "CBP-QSNN: Spiking Neural Networks Quantized Using Constrained Backpropagation"
_ICLR.cc/2023/Conference — Submitted to ICLR 2023_

### Official Review · Reviewer_6mKQ · 2022-10-20

**Confidence:** 2
**Correctness:** 3
**Technical Novelty And Significance:** 3
**Empirical Novelty And Significance:** 4
**Recommendation:** 3

**Clarity, Quality, Novelty And Reproducibility:**

The first part of the manuscript is clear enough.  The later parts do tend to resort to acronyms and technicalities that makes it difficult to read.  The tables are really not clear.  They would be better presented as graphs.
Code is provided in the supplementary material so I guess the results are reproduceable.

**Strength And Weaknesses:**

Strengths:
The method seems to work.  It is applicable to more or less any kind of SNN irrespective of how it was trained.

Weaknesses:
It is not clear to me quite how beneficial the method is given the tables.  The tables are quite difficult to read and do not always appear to compare like with like.

**Summary Of The Paper:**

The manuscript makes a case for weight quantisation being necessary in deep SNNs.  It is shown that basic quantisation can introduce noise.  The authors then describe a constrained backprop using Lagrangian functions as objectives, showing that these can be tuned to different architectures.  Experiments appear to show that the method works both absolutely and when compared to other quantisation methods.

**Summary Of The Review:**

The manuscript presents what appears to be a genuinely useful technique.  However, the theoretical justification is quite involved and the results are difficult to read.
Whilst the material is strictly in scope, my feeling is that the manuscript in general is on the edge of the technical remit for *this conference*; this is evidenced to some extent by there being only one reference to past ICLR events.
On balance, I have too many questions to err more positively.

---

### Official Review · Reviewer_k52R · 2022-10-24

**Confidence:** 4
**Clarity, Quality, Novelty And Reproducibility:** sound
**Correctness:** 3
**Technical Novelty And Significance:** 2
**Empirical Novelty And Significance:** 3
**Recommendation:** 3

**Strength And Weaknesses:**

Strength:

the paper is clearly written and easy to follow, figure 1 directly conveys the main results and the contribution of this paper.

Weaknesses:

(1) The related work is narrowed to the SNN quantization area, but there are many papers for quantization DNNs, such as BinaryConnect, BNN, XNOR[1], TWN[2]. The author has mentioned BinaryConnect, BNN in the related work and said that "such
binary SNNs are incapable of processing event-based datasets".  However, the reviewer is confused that why CBP is suitable for event-based datasets and prior works are not. Since the paper focus on the quantization of weights, please note why previous DNN quantization methods can not be applied to SNN weights.

(2) The constraint functions are quite heuristic.

For example, in figure2, how are the functions designed? for figure 2(a) with g=1, are weights larger than 1 stored as fp32? The reviewer thinks that even if the weight matrix is quantized with mostly 0 value and very sparse fp32 values, the position/index of fp32 values will occupy additional memory.

(3) The experimental results seem quite mixed, which are inconsistent with the SOTA claims in the conclusion.

For example, in Table3 CIFAR10, CBP-QSNN-STBP and STBP + ADMM have the same network architectures and similar accuracy (from 89.01 to 89.67 is no big deal for cifar10), but the memory usages are also the same, which make the reviewer very confused since the main goal of this paper is to reduce memory.

For CIFAR-10, CBP-QSNN-STBP with 63.8M weights performs worse than BinaryConnect-SNN with only 13.0M weights.

The authors are encouraged to re-organize the experimental results, with the same network arch and similar # weights, CBP has better accuracy, or with the same network arch and similar accuracy, CBP has less memory usage, etc.



[1] Rastegari M, Ordonez V, Redmon J, et al. Xnor-net: Imagenet classification using binary convolutional neural networks[C]//European conference on computer vision. Springer, Cham, 2016: 525-542.

[2] Li F, Zhang B, Liu B. Ternary weight networks[J]. arXiv preprint arXiv:1605.04711, 2016.

**Summary Of The Paper:**

The author proposes a method to quantize the weights of SNN model using constrained backpropagation (CBP) with the Lagrangian function as an objective function. CBP achieves SOTA results in terms of classification accuracy and weight-memory usage in comparison with previous methods for QSNNs on CIFRA-10/100, ImageNet, DVS128 Gesture, and CIFAR10-DVS datasets.

**Summary Of The Review:**

See above

---

### Official Review · Reviewer_4gmV · 2022-10-24

**Confidence:** 4
**Correctness:** 3
**Technical Novelty And Significance:** 1
**Empirical Novelty And Significance:** 2
**Recommendation:** 3

**Clarity, Quality, Novelty And Reproducibility:**

Clarity

• The overall process of the algorithm can be easily understood, but more details should be described especially using accurate mathematical formulas as much as possible. Examples can be seen in weaknesses above.

Quality

• Experiments on SNN weight quantization are adequate and validated with different training methods on multiple datasets. However, there are problems in the experimental setup for comparison with previous work, which makes the final results unable to truly compare the algorithms.

Novelty

• The article applied the CBP algorithm of DNN weight quantization to the SNN weight quantization. However, there is no essential difference between the DNN and SNN models for the CBP algorithm. The novelty of this article is insufficient.

Reproducibility

• The authors have provided code to be reproduced.


**Strength And Weaknesses:**

Strength

1. This work applied CBP to QSNN and achieved good performance in various datasets using several representative learning algorithms of SNN. Even if FP32 weights are quantized to binary or ternary weights, the test accuracy of CBP-QSNN can still be basically maintained or even better. This can greatly reduce memory consumption when deploying on neuromorphic chips while maintaining performance, thereby allowing deeper and wider SNN to be tried.

2. CBP has been applied on different training methods for SNN (rate code-based backprop, temporal code-based backprop and DNN-to-SNN conversion), validating it to be a general framework for QSNNs.


Weaknesses

1. In this paper, the DNN weight quantization algorithm CBP is applied in SNN weight quantization. However, after the application is changed from QDNN to QSNN, the procedures of the algorithm actually have no change at all. Although the limited memory of neuromorphic chips makes SNN weight quantization more important, the work in this paper is still not innovative enough.

2. The design of experiments to compare with previous works on QSNNs (Table 3) is unreasonable, because there is no guarantee that the variable is unique. For example, BSNN has different network structure from CBP-QSNN-(7B-Net), then the comparison between them cannot tell whose quantization algorithm is better. In the few comparisons with the same network structure, the advantages of the CBP algorithm are not obvious. For example, CBP-QSNN-STBP with ternary weights outperforms (Deng et al., 2021) by only 0.57% in accuracy. Moreover, the experimental results on ImageNet (table 1,3) are relatively poor and I don’t know why only ImageNet isn’t tested three trials to avoid the random seed effect.

3. Some parts of the text are not clearly described: the equation of the weight-constraint function cs(w), the equation of the surrogate loss function proposed by authors for weight quantization of DNN-to-SNN conversion, calculation method of weight-memory usage, what does the clip function in the pseudocode do, and so on.

4. I don't understand why the curve of test accuracy in A.3 (Figure. 3-11) always starts from a very low value, shouldn't the model has already been pre-trained when the post-training starts?


**Summary Of The Paper:**

This paper presented CBP-QSNN, which applies the CBP algorithm proposed by Kim & Jeong in a recent study of quantized DNN as a general post-training method to quantized SNN. The authors began by introducing key elements of CBP, including the Lagrangian objective function and the weight-constraint function cs(w) in it, and pseudo-LMM aiming to include non-differentiable situation. Specifically, sawtooth-shaped cs(w) is combined with an unconstrained-weight window to make weights closer to the quantized value move toward this value preferentially. Then different CBP settings for various training methods and network structures of SNN were discussed to validate that CBP can work as a general weight-quantization framework for SNN. Finally, the authors reported experiments comparing the accuracy and weight memory usage, showing that CBP can significantly reduce the weight-memory usage without reducing performance or even improving performance. However, in the experiments comparing the CBP method with the previous works on QSNN, most comparisons are invalid because of the different network structures used.

**Summary Of The Review:**

This paper applied the CBP algorithm to SNN weight quantization and validated that CBP can be used as a general post-training framework for weight quantization by testing on various datasets with different SNN training methods. However, whether it is DNN or SNN, there is not much difference between the two for the CBP algorithm, so the innovation of this paper is insufficient. And there is a problem with the experimental setup in the article. When comparing CBP-QSNN with previous QSNN methods, there is no guarantee that the network structure is the same, which makes the experimental results unconvincing. And the performance improvement in the results compared with other algorithms is not attractive enough.

---

### Decision · Program_Chairs · 2023-01-20

**Decision:**

Reject

**Justification For Why Not Higher Score:**

low scores, no rebuttal

**Justification For Why Not Lower Score:**

N/A

**Metareview: Summary, Strengths And Weaknesses:**

### Description

The paper targets the problem of learning Spiking Neural Networks with quantized weights, where activations are binary and there is a time dimension, allowing for pre-activation potentials to accumulate. The work proposes a an alternative to common practices of training using straight-through gradient heuristic for weights. The quantization of weights is enforced gradually (similar in spirit to say graduated non-convexity). Namely, for quantization constraints a specially designed barrier function is added to the objective with a fixed schedule of its shape and with the lagrange multiplier updated every epoch. The paper demonstrates that the scheme works successfully allowing to introduce weight quantization to a range of existing training methods for training SNNs and for  DNN to SNN (trained) conversion. This is a case of interest, since typically SNNs do not consider weight quantization, unlike in Quantized NNs / BNN. Therefore well-motivated.

### Decision

The training method is different from what is usually used for weights quantization, it is interesting to have it represented and see how it performs (experimental evidence). It would contribute useful knowledge to the community. However, the initial reviews were rather negative posing a lot of questions to the work. In particular, reviewers were interested how it compares to common ST-like training methods for quantized/binary NNs. The authors did not attempt rebuttal (and I agree it would be hard to change the game), therefore must reject.

### Additional feedback

Sorry, in the view of the paper state, these comments will be drafty

* cs is never defined?

* In the conversion, why not to use per-layer objectives?

* ST-like methods could be applied in all studied cases, and it would be interesting to compare to that baseline.

Many SNNs are simply SBNs with a specialized architecture: with a grid-like unrolled computational graph as in

Wu et al (2019) Direct Training for Spiking Neural Networks: Faster, Larger, Better

or SBNs with special propagation rules.
Methodologically, it would be more efficient therefore to treat all SNN methods reducible to BNN as instances of such: apply generic BBN training methods for that specialized architectures. The research on training NNs with quantized/binarized weights and activations appears to be quite ahead of SNNs.

An SNN that is not reducible to QNN/BNN but that can be trained by the proposed method was not considered / demonstrated.

* A major issue is the choice of hyperparameters: one needs to balance 3 learning rates but the procedure for choosing these hyperparameters is not described. Data split into training/ validation is not mentioned.